# Risk Factors for Early Hospital Readmission in Geriatric Patients: A Systematic Review

**DOI:** 10.3390/ijerph20031674

**Published:** 2023-01-17

**Authors:** Francesco Cilla, Ilaria Sabione, Patrizia D’Amelio

**Affiliations:** Service of Geriatric Medicine and Geriatric Rehabilitation, Centre Hospitalier Universitaire Vaudois (CHUV), Route de Mont Paisible 16, 1011 Lausanne, Switzerland

**Keywords:** geriatrics, early readmission, risk factors

## Abstract

The number of older patients is constantly growing, and early hospital readmissions in this population represent a major problem from a health, social and economic point of view. Furthermore, the early readmission rate is often used as an indicator of the quality of care. We performed a systematic review of the literature to better understand the risk factors of early readmission (30 and 90 days) in the geriatric population and to update the existing evidence on this subject. The search was carried out on the MEDLINE, EMBASE and PsycINFO databases. Three independent reviewers assessed the potential inclusion of the studies, and then each study was independently assessed by two reviewers using Joanna Briggs Institute critical appraisal tools; any discrepancies were resolved by the third reviewer. Studies that included inpatients in surgical wards were excluded. Twenty-nine studies were included in the review. Risk factors of early readmission can be classified into socio-economic factors, factors relating to the patient’s health characteristics, factors related to the use of the healthcare system and clinical factors. Among these risk factors, those linked to patient frailty play an important role, in particular malnutrition, reduced mobility, risk of falls, fatigue and functional dependence. The early identification of patients at higher risk of early readmission may allow for targeted interventions in view of discharge.

## 1. Introduction

The older population is continuously growing increasing public health challenges. According to statistics from the Report on the State of World Population, in 2050, one out of six persons in the world (one out of four in Europe and North America) will be over 65 (16%), compared to one in eleven in 2019 (9%). The number of adults aged 80 years or over is expected to triple, from 143 million in 2019 to 426 million in 2050 [1].

In this global context, the control of health costs and the efficiency of care represent two of the most important challenges for the upcoming years. For this reason, hospitals and healthcare systems have undertaken several initiatives to limit healthcare costs, including reducing the average length of stay, centralising acute care to optimise resources and reducing bed use [2].

Early unscheduled readmission represents a major problem among older adults from a health, social and financial point of view [3]. Hence, readmission rates are frequently used as indicators of the quality of hospital care [4].

According to different studies, the readmission rate 1 month after hospital discharge varies between 7.3 and 32.7% [5,6,7], depending on the population included, the geographical areas and the different departments examined [8,9]. Different studies focused on different outcomes: all readmissions [10], unplanned readmissions [11], or avoidable readmissions [12].

Potentially avoidable readmissions are a serious burden for patients and caregivers and cause increased healthcare costs [3,13,14]. The clear identification of factors predictive of early readmission may enable the implementation of appropriate interventions to obtain significant reductions in readmission rates to hospitals for older patients [15,16,17].

Indeed, in the United States, discharge plans and strict follow-up for patients at high risk of readmission are effective in reducing the rehospitalisation rate at 30 days [18,19].

Several studies have been published on this topic in the past years; the systematic review by Pedersen et al. [20] suggests that socio-demographic risk factors, as well as organisational factors and clinical factors, may be regarded as predictors of early readmissions. However, due to the high degree of heterogeneity between different studies included in the review, the authors suggest caution in the interpretation of the results [20].

The aim of this systematic review is to update the existing evidence on this topic identifying the risk factors for early readmissions in geriatric patients hospitalised in medical units.

## 2. Materials and Methods

### 2.1. Eligibility Criteria

Studies included in this systematic review answered the research question structured by the following Participants’ Intervention (PI), Exposure Comparator Outcomes (ECO) format:

Participants: older adults aged 65 years and older were readmitted to hospital within 30 or 90 days after discharge (defined as early readmission). We excluded studies on patients discharged from psychiatric, surgical, rehabilitation or palliative units and patients transferred to rehabilitation.

Interventions/exposures:-Socio-demographic characteristics and socioeconomic determinants;-Health factors;-Healthcare utilisation;-Clinical factors.

Comparator: older adults aged 65 years and older without early re-hospitalisation.

Outcomes: 30 or 90 days readmission to hospital.

### 2.2. Study Design

We included peer-reviewed observational and intervention studies, clinical trials, prospective and retrospective controlled cohort studies and case-controlled studies written in English. Only studies published in the past decade were included (from 1 January 2012 to 31 January 2022).

We excluded studies limited to patients with specific diseases such as heart attack, dementia, pneumonia etc., case reports and narrative reviews.

### 2.3. Information Source and Search Strategy

We carried out this systematic review in agreement with the Preferred Reporting Items for Systematic Reviews and Meta-Analysis (PRISMA, the PRISMA checklist is reported in Appendix B). The protocol of this study is available on the International prospective register of systematic reviews (PROSPERO, number CRD42021292496, https://www.crd.york.ac.uk/prospero/display_record.php?RecordID=292496) (accessed on 30 November 2021). The MEDLINE, EMBASE and PsycINFO databases were searched for relevant studies using the following terms: ((“Aged” [Mesh] OR “Geriatrics” [Mesh]) AND “Patient Readmission” [Mesh] AND “Risk Factors” [Mesh] NOT “Surgery” [MeSH Terms]). The search strategy is publicly available at https://pubmed.ncbi.nlm.nih.gov/?term=%28%22Aged%22%5BMesh%5D+OR+%22Geriatrics%22%5BMesh%5D%29+AND+%22Patient+Readmission%22%5BMesh%5D+AND+%22Risk+Factors%22%5BMesh%5D+NOT+Surgery%5BMeSH+Terms%5D&filter=dates.2012%2F1%2F1-2022%2F1%2F31&filter=hum_ani.humans&filter=lang.english&filter=age.aged&size=200 (accessed on 1 February 2022) for PubMed, “(‘aged’/exp OR ‘geriatrics’/exp) AND ‘patient readmission’/exp AND ‘risk factors’/exp NOT ‘surgery’/exp AND [01-01-2012]/sd NOT [01-02-2022]/sd AND [english]/lim” for EMBASE and “MeSH: geriatrics OR MeSH: old AND MeSH: patient readmission AND MeSH: risk factors NOT MeSH: surgery AND PsycInfo Classification: 2860 Gerontology for PsycINFO”.

The final search was done on 1 February 2022.

### 2.4. Study Selection

Two reviewers, working independently, screened the studies retrieved by the search according to the inclusion/exclusion criteria. Two reviewers independently evaluated the inclusion of each study; discrepancies between the two reviewers were solved by the third. The Rayyan^®^ tool (a web tool designed to help researchers working on systematic reviews available at Rayyan—Intelligent Systematic Review—Rayyan) was used to speed up the article selection process. All the papers retrieved by the search responding to inclusion/exclusion criteria were included; biases were evaluated for each article and noted in a dedicated database. In 17 studies [10,21,22,23,24,25,26,27,28,29,30,31,32,33,34,35,36], the corresponding author was contacted for further information. Only 10 authors [10,21,22,25,27,28,29,30,31,33] sent the requested information. None of the studies for which we did not receive the requested additional information were excluded; however, the quality assessment for these studies was incomplete [23,32,34,35].

### 2.5. Data Extraction and Analysis

Three thousand and twenty-one articles were retrieved by the search strategy: 1491 from EMBASE + 1410 from PubMed + 120 from PsycINFO. After removing the duplicates, we retained 2532 articles for the systematic revision. We excluded 2139 articles for violation of inclusion criteria. Hence, 393 full-text articles were reviewed as previously described. After reading the full-text article, 364 articles were excluded due to the following reasons: wrong study population (*n* = 193), wrong outcome (*n* = 121) or full paper non-available (*n* = 50). Twenty-nine articles were included in the Review (Figure 1).

From each study, 2 reviewers extracted: publication year, design of the study and analytic model, participants number, mean age, gender, number of readmitted patients, rate of readmission, duration of the follow-up, data sources, clinical setting, main outcomes, variables analysed and predictors of readmission identified.

### 2.6. Quality Assessment

The quality of the studies was assessed by the standardised Joanna Briggs Institute (JBI) critical appraisal tools [37]: JBI Critical Appraisal Checklist for case-control studies, JBI Critical Appraisal Checklist for cohort studies and JBI Critical Appraisal Checklist for quasi-experimental studies were used.

Two reviewers independently evaluated the quality of each study; discrepancies between the two reviewers were solved by discussion and consensus among reviewers or by the third reviewer. The quality across studies was assessed using a graphic representation of the percentage of studies with a strong, moderate or weak rating for each section considered in the evaluation of the quality of individual studies.

## 3. Results

### 3.1. Study Design and Characteristics

Twenty-nine studies published between 2012 and 2022 were included in the review [3,10,11,12,21,22,23,27,28,29,30,31,32,33,34,35,36,38,39,40,41,42,43,44,45,46,47,48,49]. The participants included in the different studies ranged between 111 [41] and 1,463,781 [27], with a total number of patients included in the review of 3,859,134. The majority of the included studies were retrospective cohort studies [3,12,21,27,31,33,34,36,42,43,44,46,47,48,49], nine were prospective cohort studies [10,22,23,29,30,35,38,40,41], four were case-control studies [11,28,32,45] and one was a quasi-experimental study [39]. Eighteen studies were monocentric [12,21,22,23,28,29,30,31,33,34,35,36,39,40,41,43,45,48] and 11 were multicentric [3,10,11,27,32,38,42,44,46,47,49]. Twelve were conducted in Europe [10,12,21,27,28,30,31,35,38,43,46,48], nine in the USA [22,29,32,34,40,41,42,44,49], five in Asia/Middle East [23,33,36,39,47] and three in Australia [3,11,45]. Nineteen studies [3,10,11,12,22,23,27,31,33,35,36,38,39,40,41,42,43,44,48] presented results according to multivariate analysis.

The study design of each study is detailed in Appendix A.

### 3.2. Study Population

The minimum age for patients’ inclusion differs amongst different studies; however, all the studies reported an average age of at least 65 years.

Eleven studies [3,10,11,22,29,30,41,44,45,47,48] included only patients aged 65 or older, one study [40] included patients aged 55 or older, one study [23] included patients aged 60 or older, two studies [21,38] included patients aged 70 or older and two studies [34,35] included patients aged 75 years or older.

Nine studies [12,27,28,32,33,36,42,46,49] included patients of all ages, with a mean age of 65 years. Finally, the age of inclusion is not mentioned in three studies [31,39,43]; nevertheless, patients enrolled were admitted to a geriatric ward. The characteristics of the population included in the different studies are detailed in Appendix A.

### 3.3. Risk Factors for Early Hospital Readmission

The incidence rates of readmission (both unplanned and for any cause) within 30 days ranged from 10.3% [3] to 37.6% [23], while the rates within 90 days ranged from 16% [21] to 58% [47].

Here, we take into account risk factors for readmission at 30 or 90 days according to the definition of early readmission in literature [10,12,28,30,38,42,48,50,51]. Twenty-three studies [12,22,23,27,28,29,30,31,32,33,34,35,36,39,40,41,42,43,44,45,46,48,49] investigated the risk factors of readmission at 30 days, three studies at 90 days [10,21,47] and three studies [3,11,38] at 30 and 90 days.

The risk factors for readmission highlighted by the studies included in this review can be classified into four categories: socio-demographic and socio-economic factors, health factors, healthcare utilisation and clinical factors.

Fifteen studies [10,11,12,23,29,30,31,36,38,39,41,45,46,47,49] analysed all four groups of exposure, while six studies [3,21,22,27,32,34] considered only one group of exposure.

The majority of the studies identified as risk factors for early readmission older age [3,28,33,47], male gender [3,30,46], a poor socio-economic status [3,30,40,42,45], malnutrition [21,23,38,46], multi-morbidity [11,12,27,30,31,36,39,40,45,49], liver diseases [10,46,49], heart failure [28,33,44,47,49], anaemia [36,46,48], recent hospitalisation [10,12,45] and longer hospital length of stay [12,30,36,46,49].

Different risk factors of readmission are detailed in Appendix A and Figure 2.

### 3.4. Quality Assessment

The results of the critical appraisal process are reported in Figure 3. Among the included studies, Fitriana et al. [23] and Wang–Hansen et al. [35] received the highest score (9 points), while Ben–Chetrit et al. [47] and Scott et al. [45] received the lowest score (3 points).

D1–D11 are detailed in JBI Critical appraisal checklist [37] for cohort studies [3,10,12,21,22,23,27,29,30,31,33,34,35,36,38,40,41,42,43,44,46,47,48,49], case-control studies [11,28,32,45] and quasi-experimental studies [39]).

The global quality assessment across the studies is presented in Figure 4.

We highlight that the item “Outcome measured in a standard, valid and reliable way” (D7 for Cohort studies and D8 for case-control studies and quasi-experimental studies) has a high risk of bias in the cohort studies (only 42% of the studies [10,12,23,30,35,38,40,42,48,49] have a low risk of bias) and in the case-control studies (only 50% of the studies [11,32] with a low risk of bias). This result is explained by the attribution of the “high risk of bias” category to studies that included readmissions only in the same units of index hospitalisation; this potentially leads to an underestimation of the outcomes.

Regarding the item “Follow-Up complete” (D9 for Cohort studies) in the Cohort studies, a high risk of bias was attributed if the authors did not have information on non-readmitted patients (who could potentially be dead or hospitalised elsewhere). According to these criteria, only 45% of the cohort studies [10,12,22,23,27,31,35,38,40,42,49] have a complete follow-up.

In 20% [21,28,32,34,45,49], the statistical analysis (D11 for cohort studies, D10 for case-control studies and D9 for quasi-experimental studies) was not appropriate, given the absence of multivariate analysis on risk factors for readmission. For 10% [29,46,47] is unclear if an appropriate statistical analysis was performed.

## 4. Discussion

The aim of this review is to broadly evaluate the risk factors for early hospital readmission in geriatric patients; with this aim, we included very heterogeneous studies. The studies included widely differed in design, setting and data collection methods and were carried out in different countries with different cultures, health systems and economic situations.

In addition, the definition of early readmission differs amongst the studies as readmission for any causes is included in 14 studies [10,21,27,29,30,32,33,34,35,36,43,44,47,49], only unplanned readmissions in 12 studies [3,11,22,23,28,38,39,40,41,42,45,46] and potentially avoidable readmissions in three studies [12,31,48]. Taking into account that both planned and unplanned readmission potentially overestimate the readmission rate, however, the definition chosen by different authors was not included in the quality assessment as it depends on the declared study’s outcome.

Despite this heterogeneity, this review allows us to synthesise and update the knowledge on risk factors associated with early readmissions. According to the included studies, we classified risk factors for early readmission into four categories: socio-demographic and socio-economic factors, health factors, healthcare utilisation and clinical factors related to the index admission.

The incidence rates of readmission (unplanned readmissions or any causes of readmissions) vary markedly between different studies [3,21,23,47]. These variations have already been highlighted by previous systematic reviews [8,20] and may be due to different populations, countries’ health system differences and the different definitions adopted for early readmission.

Regarding the socio-demographic and socio-economic factors, our review highlights contrasting results. Older age [3,28,33,47] and male gender [3,30,46] were found to be associated with an increased risk of early readmission in some studies. However, these findings were not confirmed by other studies, as in one study [35], female gender was found to be a risk factor for early readmission, whereas younger age was associated with early readmission by two studies [35,49]. These conflicting results could be explained by different settings as, in some countries, older patients are mostly treated at home or in nursing homes and, thus, are not transferred to an acute hospital [35].

Nevertheless, the majority of the studies did not find a significant association between gender and age and readmission rate [10,11,12,22,23,29,31,33,36,38,39,40,42,43,44,45,48]. A poor socioeconomic status (living in a disadvantaged area [42], low level of education [30], belonging to minorities [3,40], speaking a foreign language [45]) increase the risk for early readmission as well as living in a rural area [3]. These findings are similar to previous systematic reviews on this topic [8,20,53].

Regarding the health factors, factors associated with frailty syndrome (malnutrition [21,23,38,46], lower daily steps [22], cognitive impairment [28,29,30,38], multi-morbidity [11,12,27,30,31,36,39,40,45,49]) and functional dependence [31] these are associated to an increased risk of early readmission.

These findings are partially in contrast with Wang–Hansen et al. [35], who suggested that better cognitive performance is a risk factor for hospital readmission. According to these authors, this could be due to the practice, diffused in the Norwegian healthcare system, to mostly treat patients with cognitive decline in nursing homes without transferring them to acute hospitals [35].

Finally, our review found that some underlying comorbidities (chronic obstructive pulmonary disease [44,49], kidney failure [28,49], cerebrovascular diseases [28], diabetes [46,49], hypertension [44], atrial fibrillation [46], cardiovascular diseases [10], liver diseases [10,46,49] and depression [23]) are associated to increased risk of early readmission.

Regarding healthcare system utilisation, our review found that patients with a recent hospitalisation [10,12,45] or Emergency Department visits [30,43], frequent hospital visitors [33,39] or patients with longer hospital length of stay [12,30,36,46,49] are at higher risk of early readmission. These findings are similar to the ones obtained by previous systematic reviews [8,20].

Silber et al. [32] found that hospitalisation in non-teaching hospitals is associated with an increased risk of readmission. Finally, Maddox et al. [42] found that Medicaid patients have a higher readmission risk and those patients have poorer socio-economical conditions.

Regarding the clinical factors, some diagnoses made during the index admission (heart failure, exacerbation of chronic pulmonary disease, hyponatremia, pressure ulcers, anaemia and sepsis) are associated with early readmission.

Furthermore, the prescription of 15 or more drugs during the hospital stay [12], non-compliance to venous thromboembolism prophylaxis [34] and adverse clinical events during hospitalisation [10] were associated with increased risk of early readmission.

Finally, Van Seben et al. [38] focused on the period immediately following the discharge showing that the development of cognitive impairment, fatigue or falls one month after discharge is associated with an increased risk of early readmission.

As pointed out above, the contrasting results could be explained by the differences between the different countries in which the studies were carried out and the differences in healthcare systems.

Therefore, we compared the results of the different macro-areas (Europe, USA, Asia and Oceania).

From this comparison, it can be inferred that some risk factors for early readmission are common to the different macro-areas (in particular, a length of hospitalisation of at least 6 days [12,30,36,46,49] and the presence of multi-morbidities [11,12,27,30,31,36,39,40,45,49]), but some differences can be highlighted for the other risk factors.

In particular, our review highlights how social isolation and low socio-economic level represent risk factors for readmission, mainly in studies carried out in the USA [40,42] and Oceania [3,45]. It should be pointed out that in the studies carried out in Europe and Asia, this risk factor was less frequently evaluated.

Another difference between the various macro-areas concerns the diagnosis of dementia, which represents a risk factor for readmission in Europe and Israel [28,30,38,46,47], but not in Asia [11] and only for patients residing in nursing homes in the USA [40,44].

Furthermore, the presence of cancer is a risk factor for readmission in some studies carried out in Europe and Asia [12,36,46], whereas this factor is not studied in the papers carried out in the USA [22,29,32,34,40,41,42,44,49].

Finally, this review shows that malnutrition [21,23,38,46], male gender [3,30,46] and age [3,28,33,47] are among the risk factors most emphasised among the included studies but are not present in any of the studies carried out in the USA [22,29,32,34,40,41,42,44,49].

These differences are probably due to cultural, environmental and health systems differences amongst the different countries.

The strength of this review is the strict methodology applied. Specifically, the use of a standardised tool (JBI tool) to assess the bias of the included papers allow us to standardise the evaluation and thus to draw objective conclusions even in the case of heterogeneous papers. However, due to the high heterogeneity of the different studies, the results are difficult to compare. Furthermore, all the included studies have biases according to the JBI tool.

It is noteworthy to underline that about half of the studies included evaluated early readmission in a single ward or hospital; this probably leads to an underestimation of the total readmission rate. This bias is not negligible as, according to the literature [54], the rate of readmissions to a different ward or hospital varies between 20 and 40%.

Another limit of this review is the type of readmission included in different studies, as about half of the studies [10,21,27,29,30,32,33,34,35,36,43,44,47,49] are not limited to unplanned readmissions; this bias can overestimate the early readmission rate.

Due to these limitations, the authors believe that there is still room for further experimental studies with clear outcomes on early, unplanned hospital readmission in acute geriatric patients.

The quality assessment carried out during the review will be particularly important for researchers wishing to carry out future work on the subject of readmissions, as it will enable them to have a more rigorous methodology with a lower risk of bias.

## 5. Conclusions

Our review gives a broad overview of risk factors for early hospital readmission in geriatric patients. The early identification of the patients at higher risk of early readmission may allow for planning targeted interventions in view of hospital discharge.

Amongst the risk factors highlighted by different studies, those associated with frailty syndrome play an important role, in particular malnutrition, reduced mobility, risk of falls, fatigue and functional dependence. This result is of paramount importance as these conditions are modifiable and must be taken into account in patients’ evaluation and treatment. Careful nutritional management and early mobilisation during hospitalisation are, therefore, low-cost measures that can lead to a reduction in the readmission rate [21,22,23,38,41] with significant improvement in patients’ quality of life and reduction of health costs.

## Figures and Tables

**Figure 1 ijerph-20-01674-f001:**
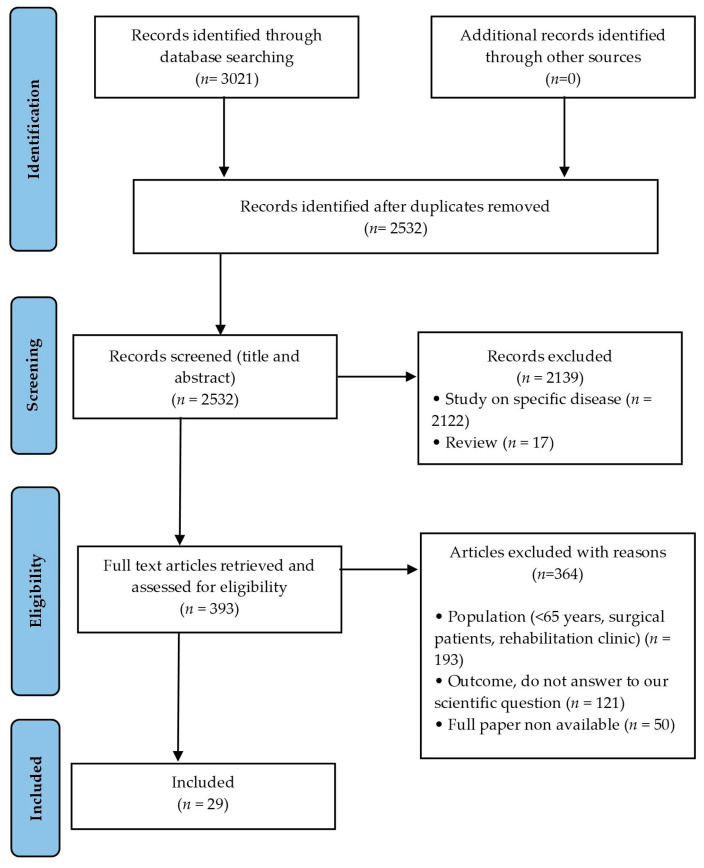
Flowchart of the study selection process.

**Figure 2 ijerph-20-01674-f002:**
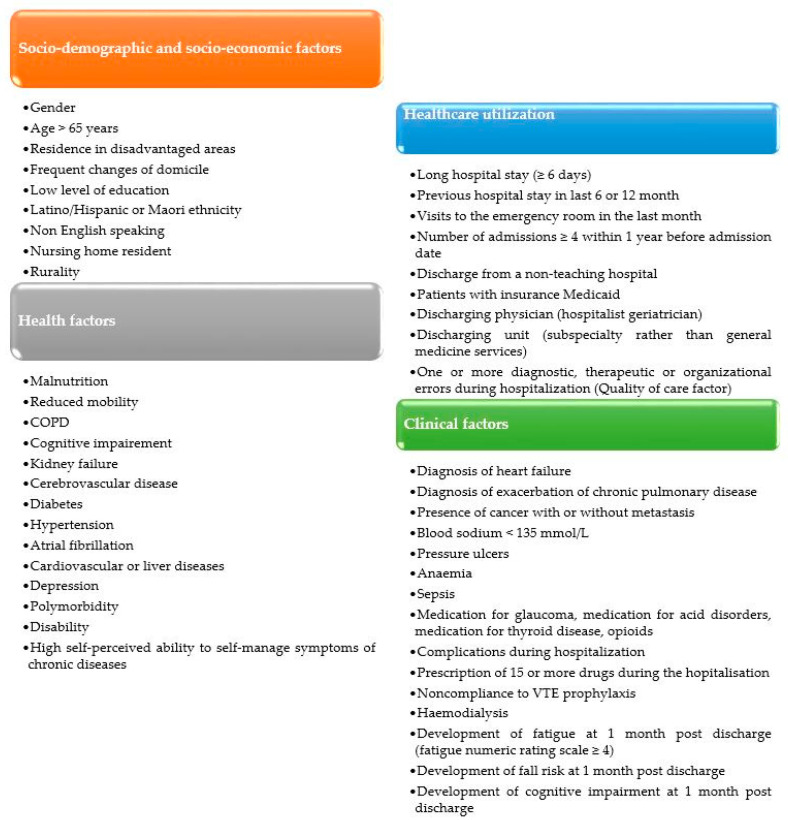
Risk factors for early hospital readmission.

**Figure 3 ijerph-20-01674-f003:**
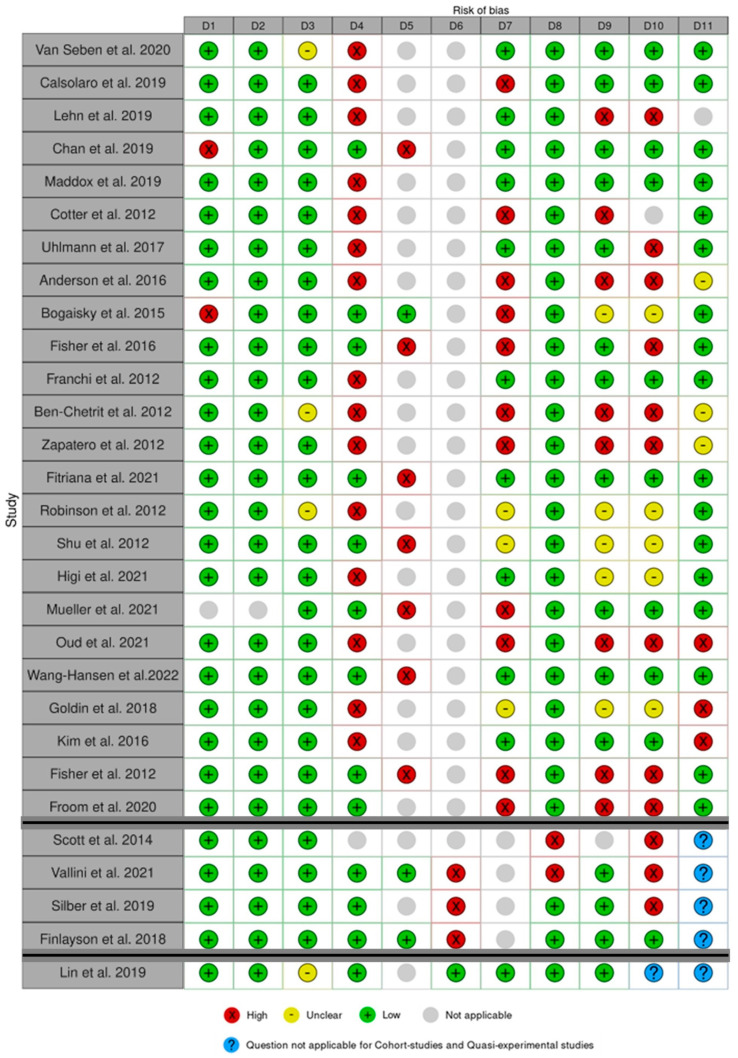
Quality assessment for cohort studies in the first part, case-control studies in the second part and quasi-experimental studies in the third part. The Robvis Tool^®^ [52] was used for this figure. Van Seben et al. 2020 [38], Calsolaro et al. 2019 [31], Lehn et al. 2019 [30], Chan et al. 2019 [40], Maddox et al. 2019 [42], Cotter et al. 2012 [43], Uhlmann et al. 2017 [12], Anderson et al. 2016 [29], Bogaisky et al. 2015 [44], Fisher et al. 2016 [22], Franchi et al. 2012 [10], Ben-Chetrit et al. 2012 [47], Zapatero et al. 2012 [46], Fitriana et al. 2021 [23], Robinson et al. 2012 [3], Shu et al. 2012 [36], Higi et al. 2021 [48], Mueller et al. 2021 [27], Oud et al. 2021 [21], Wang-Hansen et al. 2022 [35], Goldin et al. 2018 [34], Kim et al. 2016 [49], Fisher et al. 2012 [41], Froom et al. 2020 [33], Scott et al. 2014 [45], Vallini et al. 2021 [28], Silber et al. 2019 [32], Finlayson et al. 2018 [11], Lin et al. 2019 [39].

**Figure 4 ijerph-20-01674-f004:**
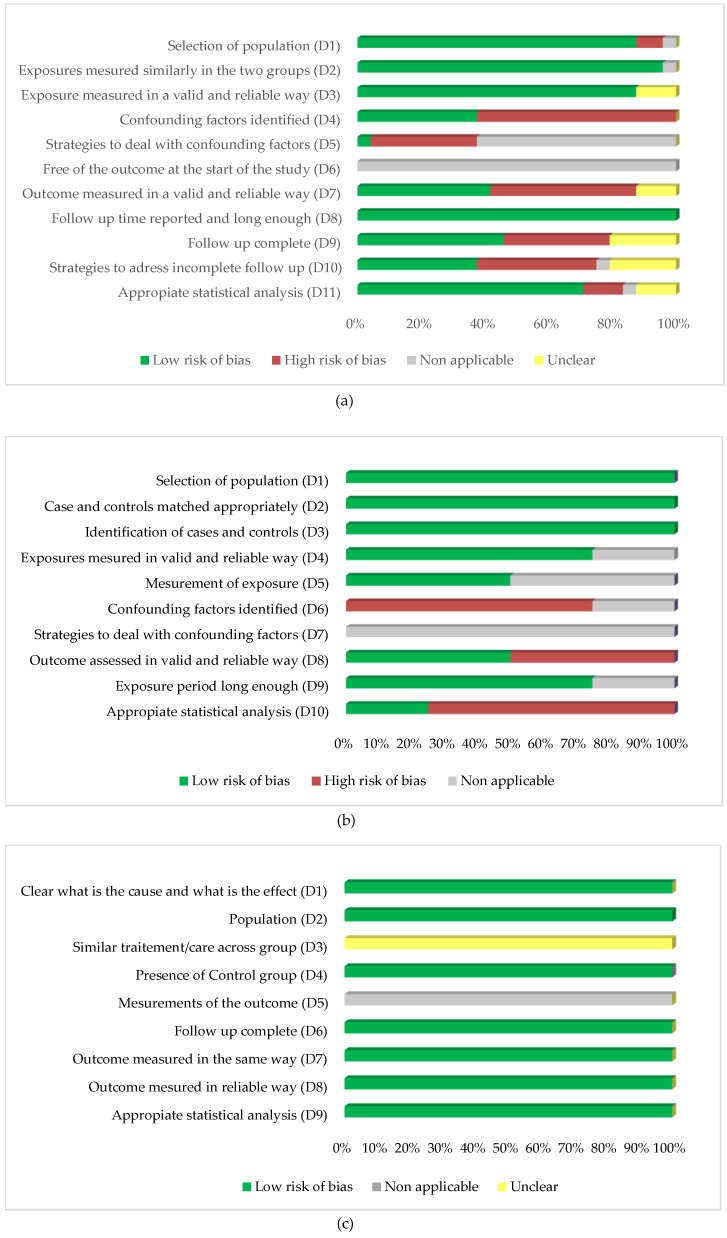
The global quality assessment across the studies for cohort studies (**a**), case-control studies (**b**), and quasi-experimental studies (**c**).

## Data Availability

All the available data are available to the public and presented in the manuscript.

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
