# Peer review of "Risk Factors for Early Hospital Readmission in Geriatric Patients: A Systematic Review"

_ijerph, 2023, doi:10.3390/ijerph20031674_

Round 1

Reviewer 1 Report

The structure of the paper needs to be refined. The numerical values are calculated unreliably. Graphs and tables are illegible.

1. Table 1 is completely illegible. Please describe the content of the table in the next 29 points of work, each of which will consist of points, containing content from individual columns of this table.

2. In line 194 and in Figure 3, the designations D1-D11 appear. These designations should be used consistently in all the work. A similar remark applies to the other forms included in the appendix.

3. Panel (a) in Figure 4 has 11 bars and only 6 captions. The descriptions in this panel are not consistent with the appendix and notations in Figure 3. Reviewer guess is the first bar is D1, the next is D3, then D5, D7, D8, and D11. Why these and in that order?

4. Similar remarks to the other panels in Figure 4.

5. in Row 202 for D7 there is a value of 42%, which is supposed to be the result of calculations10/29 (10 papers listed in row 203,  work 39, which can be found in Figure 3, is missing there). In line 201, it should be written that the calculations refer to D7 from Figure 3.

6. Other values are also difficult to interpret in the context of the data presented in Figure 3.

Reviewer 2 Report

Thank you for an interesting study. I hope this few attached comments may be to any use for you. 

Reviewer 3 Report

The authors chose to do a review of existing studies regarding hospital readmission, which is an important topic. The materials and method are sufficiently described, and the findings are solid. The findings are well-categorized and explained. 

However, something currently missing in this paper, which is expected to be quality research, is the significance and methodological implications from the existing beyond otherwise just a summary of their findings. From my perspective, the authors should address/mention the following two questions:

1. Please highlight the methodological advancement of your research compared to previous reviews on similar topics. What does this new method bring you? You mentioned heterogeneity, but this is not a direct manifestation of your contribution. Does it bring you more research that corroborates certain findings, or certain contests and debates that require more scholarly attention?

2. While you have summarized certain factors that could affect readmission, I did not see further comments on areas of current studies that need improvement. For example, beyond summarizing findings, you should also comment on methods of existing studies. Could the methods lead to certain limitations that affect their findings? This is to encourage critical reflection in your review that you are not simply taking others' findings as granted. You have done it well regarding the conflicting results from age - as different institutional setting prefers elderlies to receive care at home. However, the rest of your discussion is plain. I cannot see what new insights are provided compared to earlier review studies, nor what areas you think future studies should pay attention to or try to improve upon. 

Another comment on countries. You have realized that different healthcare systems will lead to different readmission rates. Your study has heterogeneous samples, so you should also try to comment more on healthcare systems right? What differences are these studies suggesting by, for example, comparing studies done in Europe and the ones done in North America, or in Scandinavia vs. West Europe?

These are the comments I think the authors should address before this paper is ready for publication. 

Round 2

Reviewer 1 Report

All the reviewer's comments were included in the revised paper. The readability of the article and the presented argumentation has significantly improved.

Reviewer 3 Report

The authors have addressed the raised questions. The paper is ready for publication.